# Towards Data-Driven Offline Simulations
# for Online Reinforcement Learning

**Shengpu Tang**[1*†], **Felipe Vieira Frujeri**[2], **Dipendra Misra**[3],
**Alex Lamb**[3], **John Langford**[3], **Paul Mineiro**[3], **Sebastian Kochman**[2†]

[1]University of Michigan, Ann Arbor; [2]Microsoft Azure AI; [3]Microsoft Research NYC
[*]Work done during internship at Microsoft; [†]Corresponding authors

shengpu.tang@outlook.com, {fevieira,dimisra,lambalex,jcl,pmineiro,sebastko}@microsoft.com

## Abstract

Modern decision-making systems, from robots to web recommendation engines, are expected to adapt: to user preferences, changing circumstances or even new tasks. Yet, it is still uncommon to deploy a dynamically learning agent (rather than a fixed policy) to a production system, as it's perceived as unsafe. Using historical data to reason about learning algorithms, similar to offline policy evaluation (OPE) applied to fixed policies, could help practitioners evaluate and ultimately deploy such adaptive agents to production. In this work, we formalize offline learner simulation (OLS) for reinforcement learning (RL) and propose a novel evaluation protocol that measures both fidelity and efficiency of the simulation. For environments with complex high-dimensional observations, we propose a semi-parametric approach that leverages recent advances in latent state discovery in order to achieve accurate and efficient offline simulations. In preliminary experiments, we show the advantage of our approach compared to fully non-parametric baselines. The code to reproduce these experiments will be made available at https://github.com/microsoft/rl-offline-simulation.

## 1 Introduction & Related Work

Exploration is one of the central problems of RL and has been studied primarily assuming access to an online environment (Sutton & Barto, 2018). Offline RL, the problem of learning a policy from a previously collected experience history, has gained a lot of attention in the recent years (Levine et al., 2020; Fu et al., 2020). Yet the combination of the two, i.e., reasoning about exploration and agent's learning process given just an offline dataset, though a problem of tremendous potential value, has not been covered extensively by the RL research community. *Offline policy evaluation* (OPE), a subproblem of offline RL, focuses on evaluating performance of a fixed policy using an offline dataset (Figure 1). In many real-world scenarios, including recommender systems, personalizable web services, robots required to adapt to new tasks, etc., instead of having fixed policies, we would like the agent to continue learning after deployment. This requires the agent to explore and react to its experience in the environment by adapting its policy. OPE ignores these factors and is not the right framework to assess such agents. In this work we propose *offline learner simulation* (OLS) as a way to evaluate non-stationary agents given just an offline dataset.

A natural approach to simulate learners can be to leverage model-based RL (also used successfully as an OPE method - see Fu et al. (2021)) - by learning a world model on the offline dataset, and then using that model to generate new rollouts. While simple to use, the learned model incurs bias that may be hard to measure and reason about. On the other side of the spectrum, we have non-parametric approaches replaying data in certain ways to match the true environment's distribution: Li et al. (2011) discussed this approach in the contextual bandit setting, and Mandel et al. (2016) extended the idea to Markov decision processes (MDPs). These methods are provably unbiased and allow for

simulating a learning process with provable absolute accuracy, but become inefficient for all but the simplest toy problems. More realistic environments, with rich observations and stochastic transitions, would lead to simulation terminating after few steps, deeming these methods impractical.

In this work, we incorporate recent advances in latent state discovery (Du et al., 2019; Misra et al., 2020; Lamb et al., 2022), which allow one to recover the unobserved latent states from potentially high-dimensional rich observations, with the model-free data-driven approaches proposed in Mandel et al. (2016), to improve their efficiency and practicality. In the preliminary experiments, we evaluate the methods by comparing the obtained simulations to the true online learners in terms of fidelity and efficiency. We show that newly proposed methods are able to simulate a learning process with high fidelity, and are capable of producing longer simulations than fully non-parametric approaches.

## 2 Problem Setup

We consider reinforcement learning (RL) in block Markov decision processes (Block MDPs), defined by a large (possibly infinite) observation space $\mathcal{X}$, a finite unobservable state space $\mathcal{S}$, a finite action space $\mathcal{A}$, transition function $p : \mathcal{S} \times \mathcal{A} \to \Delta(\mathcal{S})$, emission function $q : \mathcal{S} \to \Delta(\mathcal{X})$, reward function $r : \mathcal{X} \times \mathcal{A} \times \mathcal{X} \to \Delta(\mathbb{R})$, initial state distribution $\mu_0 \in \Delta(\mathcal{S})$, and discount factor $\gamma \in [0, 1]$. A policy $\pi : \mathcal{X} \to \Delta(\mathcal{A})$ specifies a distribution over actions for each observation. In RL, a *learner* (often realized by executing a learning algorithm), denoted by $\mathbb{A}$, defines a mapping from some history of interactions of arbitrary length $\tau \in \mathcal{H}$ to a policy $\pi_\tau : \mathcal{X} \to \Delta(\mathcal{A})$. Here, a history of interactions of length $t$ is an ordered sequence of transition tuples $\tau_{1:t} = [(x_{t'}, a_{t'}, r_{t'}, x'_{t'})]_{t'=1}^t \in \mathcal{H}_t$, and $\mathcal{H} = \mathcal{H}_0 \cup \mathcal{H}_1 \cup \cdots$ is the set of all histories. Consider the interaction cycle between the learner and the environment: at step $t$, the learner has seen its interactions with the environment during steps $t' = 1 \ldots t$ and makes use of the history so far $\tau_{1:t}$ to define its policy $\pi_t = \pi_{\tau_{1:t}}$ for subsequent interaction(s). Overall, the learner follows a non-stationary policy where, importantly, the sequence of policies $\{\pi_1, \ldots, \pi_t\}$ that constitute this non-stationary policy is not known in advance.

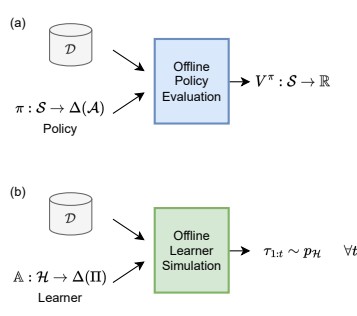

(a) $\mathcal{D}$ → Offline Policy Evaluation → $V^\pi : \mathcal{S} \to \mathbb{R}$

$\pi : \mathcal{S} \to \Delta(\mathcal{A})$ Policy

(b) $\mathcal{D}$ → Offline Learner Simulation → $\tau_{1:t} \sim p_\mathcal{H}$ $\forall t$

$\mathbb{A} : \mathcal{H} \to \Delta(\Pi)$ Learner

Fig. 1: OPE vs OLS

In this work, we consider the problem of offline learner simulation (OLS), which is used to gain an understanding of what a learner would "perform" in the real environment; we might be interested in how the learner would gather data and explore the environment, or how quickly the learner would converge to the optimal policy. Given a logged dataset of past interactions $\mathcal{D} = \{x_i, a_i, r_i, x'_i\}_{i=1}^n$, in order to simulate a (black-box) learner up to step $T$, it is necessary to provide the learner with a history $\tau_{1:T} = [(x_t, a_t, r_t, x'_t)]_{t=1}^T \sim p_\mathcal{H}^{\text{sim}}$ that is drawn from a distribution identical to the one observed if the learner interacted with the real environment $p_\mathcal{H}^{\text{real}}$. This is in contrast to OPE, where it is usually sufficient to obtain a value function estimate (Figure 1).

### 2.1 Evaluation Protocol

A "good" offline simulation should run as accurately as possible for as long as possible. Therefore, we propose to quantify the success of offline simulations via two aspects - efficiency and fidelity.

Efficiency can be measured by the length of histories generated by the simulation before it terminates. While some simulation approaches allow the learner to run indefinitely, this often comes at the cost of large biases. Therefore, we also consider simulation approaches that have the option to "terminate". In Figure 2, sim2 is more efficient than sim1 because it terminates after more simulation steps.

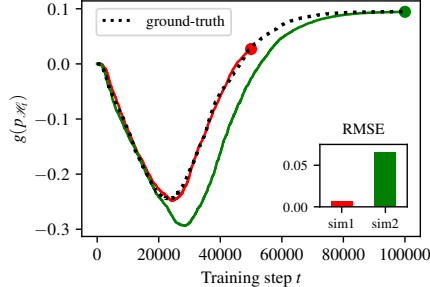

Figure 2: Example of the ground-truth learning curve as well as two offline simulations (details of this experiment are in Appendix B.1).

To measure fidelity, in theory, we want to compare the distribution of histories generated by the simulation $p_\mathcal{H}^{\text{sim}}$ to the real distribution $p_\mathcal{H}^{\text{real}}$. Since these distributions may be difficult to represent analytically, in practice, we instead use aggregate scalar statistics $g(p_\mathcal{H})$ as proxy measures.[1] For

---

[1] We caution that $g(p_\mathcal{H}^{\text{sim}})$ being close to $g(p_\mathcal{H}^{\text{real}})$ is only a necessary condition for the $p_\mathcal{H}^{\text{sim}}$ being close to $p_\mathcal{H}^{\text{real}}$, but not a sufficient condition.

example, we may have $g(p_{\mathcal{H}_t}) = \mathbb{E}_{\tau_{1:t} \sim \mathcal{H}_t}[V(\mathbb{A}(\tau_{1:t}))]$ - the expected policy performance after the learner has received a length-$t$ history $\tau_{1:t} \sim p_{\mathcal{H}_t}$. Note that the specific choice of $g$ is dependent on the learner and the environment; some alternative choices include: state visitation distribution in the history $\tau_{1:t}$, or the learner's model parameters. The expectation over distributions of histories can be approximated empirically using averaged results from multiple simulation runs. Suppose we are interested in the fidelity of a simulation from training step 1 to $T$. As shown in Figure 2, we can visualize $g(p_{\mathcal{H}_t})$ for $t = 1 \cdots T$ as learning curves, and measure the error in simulation as the RMSE between the learning curves from the simulation vs the ground-truth over all steps (alternatively, one may use the mean/max absolute error). For $T = 50000$, sim1 is a more "accurate" simulation than sim2, even though sim2 eventually converged to the correct policy after $T = 50000$ whereas sim1 is not able to run till convergence. Comparisons of fidelity are only meaningful for the same $T$.

Both efficiency and fidelity are important for offline simulation, yet it is not straightforward to define a single metric that captures both. Since there is usually a trade-off between the two (similar to the bias-variance trade-off), in our experiments below, we consider these two aspects separately.

## 3 Non-parametric & Semi-parametric Offline Simulation

Mandel et al. (2016) proposed several non-parametric approaches for OLS in RL including the queue-based evaluator (QBE) and per-state rejection sampling (PSRS). We focus on PSRS because it was shown to be more efficient than QBE. In PSRS, each transition in the logged dataset is only considered once, and is either accepted and given to the learner, or rejected and discarded. This ensures the unbiasedness of the overall simulation. In Appendix A.1, we restate these two algorithms with a few modifications to allow for more general settings such as episodic problems with multiple initial states and non-stationary logging policies.

A key step in PSRS (as the name suggests) is line 3 (Alg. 3) which groups transitions in the logged dataset into queues based on the from-state, and the simulation terminates whenever it hits an empty queue. This works reasonably well for tabular MDPs. For block MDPs, a naive approach is to treat the observations as the keys to group transitions by. Since we often have an infinite observation space, every observation is only seen in the logged dataset once, making this approach very inefficient and impractical because the queues would be empty before we are able to simulate long enough. Fortunately, recent advances in latent state discovery (Du et al., 2019; Misra et al., 2020; Lamb et al., 2022) allow one to recover the unobserved latent states from potentially high-dimensional rich observations. For simulating such block MDPs, we propose to first learn a latent state encoder, preprocess the logged dataset into latent states, and perform PSRS while grouping transitions using the latent states (high-level pseudo code shown in Algorithm 1; for more details, see Appendix A.2). Importantly, the simulation interfaces with learners using raw observations and is thus compatible with the learners that would be used with the original block MDP.

## 4 Proof of Concept Experiments

In this section, we conducted empirical evaluations of offline simulation for simple block MDPs. First, in a problem with known latent states, we show that using the latent states for offline simulation is more efficient than using the observations, without sacrificing simulation fidelity. Building on this result, we then consider a more challenging scenario where the latent states are not known and must be discovered from data, and demonstrate the effectiveness of our semi-parametric simulation approach even when the learned latent state encoder is not perfect. As the building block for our experiments, we introduce a $5 \times 5$ grid world navigation task (Figure 3-left) modified from Zintgraf et al. (2019), further described in Appendix B.

**Grid-World with Discrete Observations.** In this setting, we use a discrete observation space $\mathcal{X} = \mathcal{S} \times \{0, 1\}^k$ induced by the emission function $x = q(s) = s + (b_1 + \cdots + b_k 2^{k-1})|\mathcal{S}|$, where an observation is made up of the underlying state and $k$ bits $b_1 \ldots b_k$ that are stochastically sampled at every time step. We first collected 1,000 episodes following a uniformly random policy, and then used PSRS on the logged dataset to simulate tabular Q-learning. We compared PSRS with the observations $x$ vs PSRS with the underlying states $s$, and in each case, we repeated the simulation procedure for 100 runs with different random seeds. We show results for $k = 4$, where the observation space is 16 times larger than the state space. Figure 3-middle shows the learning curves for all runs, where we track the estimated value of the initial state as the $g$ function, since we are in the tabular setting and have transparency over the learner's internal parameters. In Figure 3-right we show the aggregate result for fidelity and efficiency. Both are accurate simulations as their learning curves are both

**Algorithm 1** OLS using Latent Per-State Rejection Sampling: high-level pseudo code

1: **Input:** Logged dataset $\mathcal{D} = \{(x_i, a_i, r_i, x_i')\}$ recorded by policy $\pi_b$, initial observation $x_0$
2: **Input:** Learner $\mathbb{A}$ that maps from history $\tau$ to policy $\pi$
3: **Input:** Encoder $f$ that maps from observation $x$ to latent state $z$
4: **Preprocess:** Calculate $z_i = f(x_i)$ for all $x_i$
5: **Initialize** queues[$z$], $\forall z$: group transitions from $\mathcal{D}$ by latent state $z$, into randomized queues
     // Start simulation
6: **Initialize** $\tau = [\ ]$
7: $x = x_0$                                        ▷ initial observation
8: **for** step $t = 1$ to $\infty$ **do**
9:      $\pi = \mathbb{A}(\tau)$                            ▷ update learner with the history
10:     $z = f(x)$                                 ▷ encode observation
11:      **while** no transition has been accepted for this step **do**
12:          **if** queues[$z$] is empty **then** terminate
13:          Sample a transition from queues[$z$]
14:          Perform rejection sampling      ▷ accept or reject the transition tuple based on similarity
                                                     between action distributions of the current policy $\pi$
                                                     and behavior policy $\pi_b$, given observation $x$
15:      $\tau$.append($x, a, r, x'$)                      ▷ update history with new transition
16:      **if** episode ends **then**
17:          $x = x_0$                                  ▷ start new episode
18:      **else**
19:          $x = x'$
20: **Output:** History $\tau$

Figure 3: Grid-world with discrete observations. **Left**: latent state space and action space. Observations consist of the latent state concatenated with 4 random bits. **Middle**: estimated value of starting state for real Q-learning and two simulations (using observations and using latent states directly). **Right**: fidelity and efficiency of the simulations. Both simulations have perfect fidelity, but using latent states allows for simulating longer.

overlapping exactly with real Q-learning, but using latent states is more efficient than using raw observations and led to simulations about twice as long for this problem. Additional variations of this experiment are explored further in Appendix B.2.

**Grid-World with Continuous Observations.** Next, we consider a more complex observation space, where the observation $x$ is a randomly sampled 2D coordinate within the grid cell of state $s$ (normalized to be between 0 and 1). We similarly collected the logged dataset using a uniformly random policy, leading to the state visitation distribution shown in Figure 4a. Note that since the observation space is now continuous – essentially no two $x$'s are the same – PSRS on the observation space is no longer practical. Therefore, we trained a neural network encoder for discrete latent states for kinematic inseparability abstraction, using the contrastive estimation objective similar to Misra et al. (2020). While we used a latent dimension of 50, it ended up learning only 20 discrete latent states as shown in Figure 4b. We subsequently used the learned state encoder in offline simulation of a PPO agent using PSRS and compared to several baselines. We visualized the learning curves (Figure 5) by tracking policy performance within each training epoch (Figure 5-left) and

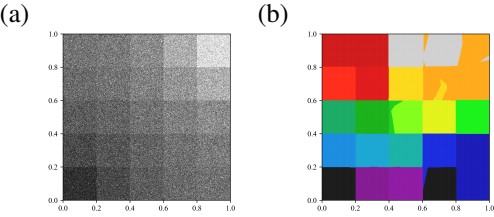

| (a)  (b) | Simulation | Efficiency (↑) | Fidelity (↓) |
|---|---|---|---|
|  | Real | $\infty$ | 0 |
|  | PSRS-oracle | 17 | 0.173 |
|  | PSRS-encoder | 16 | 0.203 |
|  | PSRS-obs-only | 50 | 1.148 |
|  | PSRS-act-only | 50 | 1.221 |
|  | PSRS-random | 50 | 1.514 |

Figure 4: Grid world with continuous observations: experiment results. **Left**: state visitation distribution and learned latent states. **Right**: quantitative results comparing efficiency (median simulation length in epochs, higher is better) and fidelity (RMSE of validation performance curves compared to PPO in a real environment, lower is better).

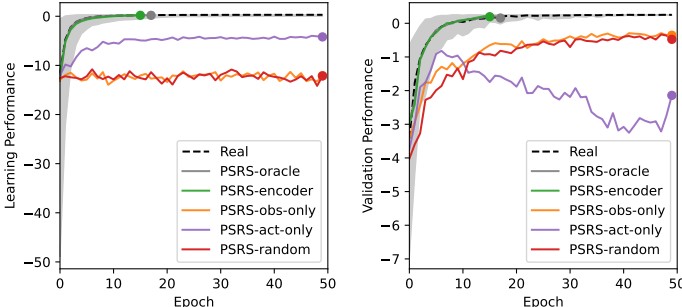

Figure 5: Learning curves of a PPO agent in grid world with continuous observations: "Real" and various PSRS simulations. **Left**: learning performance, i.e., average episode return *within* each training epoch. **Right**: validation performance, i.e., average episode return as measured in a real validation environment, *after* each training epoch. All results are averaged over 10 runs. OLS using latent PSRS, denoted here as "PSRS-encoder" faithfully reconstructs learning curves of the real, online PPO agent.

in a validation environment at the end of each epoch, obtained via averaging the returns from 10 Monte-Carlo rollouts in the true environment (Figure 5-right). Summarized numerical results on efficiency and fidelity of the validation performance are shown in Figure 4-right. Despite the errors in the latent states from the learned encoder, it performs close to the oracle encoder, and both are close to the online PPO in the real environment. All other baselines, despite being efficient, are far from accurate simulations and have non-negligible error. Overall, this experiment shows promise that for block MDPs, we can learn to encode the observations into latent states and then do offline simulations in an efficient and accurate manner, completely using offline data. Further experimental details, including explanation of the used baselines, are in Appendix B.3.

## 5 Conclusion

In this work, we studied offline learner simulation (OLS), which allows one to evaluate an agent's learning process using offline datasets. We formally described the evaluation protocol for OLS in terms of efficiency and fidelity, and proposed semi-parametric simulation approaches to handle block MDPs with rich observations by extending existing non-parametric approaches and leveraging recent advances in latent state discovery. Through preliminary experiments, we show the advantage of this approach even when the learned latent states are not perfectly correct. Code to reproduce experimental results will be released publicly upon publication of this paper.

Besides applications in recommender systems and robotics, OLS may be especially useful for multi-task and meta-learning settings, where simulation on a subset of tasks may inform us about future adaptive performance on other tasks. It may prove to be a crucial component to succeed in offline meta-RL (Dorfman et al., 2021; Mitchell et al., 2021; Pong et al., 2022). Future work should also consider removing the assumption on discrete latent topology and accounting for exogenous processes (further discussed in Appendix B.4) to handle a wider class of problems.

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

# A Algorithms

## A.1 Non-parametric Simulation for Tabular MDPs

The QBE (Algorithm 2) and PSRS (Algorithm 3) algorithms are based on Mandel et al. (2016). We made the following two modifications to the versions originally proposed to allow for more general settings such as episodic problems with multiple initial states and non-stationary logging policies.

- L4 and L6 in both algorithms, L13-15 in QBE and L16-18 in PSRS: to support episodic settings, we initialize a set of possible starting states in the init_queue, and during simulation we obtain a new starting state from this set whenever the episode ends.
- L11-12 in PSRS: to support non-stationary logging policies, assuming the logged dataset have stored the action distribution of the behavior policy that each observed action is drawn from, we recalculate $M$ for each candidate transition in the while loop.

---

**Algorithm 2** Tabular Queue-Based Evaluator (modified from Alg 1 in Mandel et al. (2016))

---

1: **Input:** Logged dataset $\mathcal{D} = \{(s_i, a_i, r_i, s_i')\}$
2: **Input:** Learner $\mathbb{A}$ that maps from history $\tau$ to policy $\pi$
3: **Initialize** queues$[s, a]$ = Queue(RandomOrder$((s_i, a_i, r_i, s_i') \in \mathcal{D}$ s.t. $s_i = s$ and $a_i = a))$, $\forall s \in \mathcal{S}, a \in \mathcal{A}$
4: **Initialize** init_queue = Queue(RandomOrder$(s_i \in \mathcal{D}$ if $s_i$ is a starting state))
   // Start simulation
5: **Initialize** $\tau = [\,]$
6: $s$ = init_queue.pop()
7: **for** step $t = 1$ to $\infty$ **do**
8: $\quad \pi = \mathbb{A}(\tau)$                    $\triangleright$ update learner with the history
9: $\quad a \sim \pi(\cdot|s)$             $\triangleright$ take an action according to the current policy
10: $\quad$ **if** queues$[s, a]$ is empty **then** terminate
11: $\quad$ Sample a transition $(\cdot, \cdot, r, s')$ = queues$[s, a]$.pop()
12: $\quad \tau$.append$(s, a, r, s')$; $\ s = s'$             $\triangleright$ update history with new transition
13: $\quad$ **if** episode ends **then**
14: $\quad\quad$ **if** init_queue is empty **then** terminate
15: $\quad\quad$ $s$ = init_queue.pop()
16: **Output:** History $\tau$

---

---

**Algorithm 3** Tabular Per-State Rejection Sampling (modified from Alg 2 in Mandel et al. (2016))

---

1: **Input:** Logged dataset $\mathcal{D} = \{(s_i, a_i, r_i, s_i', \pi_i(\cdot|s_i))\}$ where $a_i \sim \pi_i(\cdot|s_i)$
2: **Input:** Learner $\mathbb{A}$ that maps from history $\tau$ to policy $\pi$
3: **Initialize** queues$[s]$ = Queue(RandomOrder$((s_i, a_i, r_i, s_i', \pi_i(\cdot|s_i)) \in \mathcal{D}$ s.t. $s_i = s))$, $\forall s \in \mathcal{S}$
4: **Initialize** init_queue = Queue(RandomOrder$(s_i \in \mathcal{D}$ if $s_i$ is a starting state))
   // Start simulation
5: **Initialize** $\tau = [\,]$
6: $s$ = init_queue.pop()
7: **for** step $t = 1$ to $\infty$ **do**
8: $\quad \pi = \mathbb{A}(\tau)$                   $\triangleright$ update learner with the history
9: $\quad$ **while** no transition has been accepted for this step **do**
10: $\quad\quad$ **if** queues$[s]$ is empty **then** terminate
11: $\quad\quad$ Sample a transition $(\cdot, a, r, s', \pi_b(\cdot|s))$ = queues$[s]$.pop()
12: $\quad\quad$ Calculate $M = \max_{\tilde{a}} \frac{\pi(\tilde{a}|s)}{\pi_b(\tilde{a}|s)}$
13: $\quad\quad$ Sample $u \sim \text{Unif}(0, 1)$
14: $\quad\quad$ **if** $u > \frac{\pi(a|s)}{M\pi_b(a|s)}$ **then** reject the sampled transition
15: $\quad \tau$.append$(s, a, r, s')$; $\ s = s'$          $\triangleright$ update history with new transition
16: $\quad$ **if** episode ends **then**
17: $\quad\quad$ **if** init_queue is empty **then** terminate
18: $\quad\quad$ $s$ = init_queue.pop()
19: **Output:** History $\tau$

---

## A.2 Semi-parametric Simulation for Block MDPs

In this appendix section we present a more detailed version of Algorithm 1 from the main paper.

We extend Algorithm 3 to support block MDPs by assuming access to an encoder function that maps from observation $x$ to its corresponding latent state $s$ (recall that under the definition of block MDPs, there is a unique latent state $s$ that can emit given observation $x$). Note that the same extension can be applied to Algorithm 2 but is omitted here. The main parts of Algorithm 4 follows the vanilla PSRS algorithm, with the following key differences (highlighted in magenta):

- L3: the algorithm requires the latent state encoder $f$ as an additional input.
- L4: we preprocess the observations into corresponding latent states for all transitions in the logged dataset.
- L5: when grouping transitions into queues, we use the pre-computed latent states as the key instead of the observations.
- L6: the set of starting observations are taken as is without being converted to latent states.
- L12-L14: when retrieving from queues, we first compute the latent state $z$ of the current observation $x$, and then query the corresponding queue for $z$.
- L15-L18: the policies still use observations as input, maintaining the agent-environment interface of the original block-MDP that this algorithm is simulating.

---

**Algorithm 4** Latent Per-State Rejection Sampling

---

1: **Input:** Logged dataset $\mathcal{D} = \{(x_i, a_i, r_i, x_i', \pi_i(\cdot|x_i))\}$ where $a_i \sim \pi_i(\cdot|x_i)$
2: **Input:** Learner $\mathbb{A}$ that maps from history $\tau$ to policy $\pi$
3: **Input:** Encoder $f$ that maps from observation $x$ to latent state $z$
4: **Preprocess:** Calculate $z_i = f(x_i)$ for all $x_i$
5: **Initialize** queues[$z$] = Queue(RandomOrder($(x_i, a_i, r_i, x_i', \pi_i(\cdot|x_i)) \in \mathcal{D}$ s.t. $f(x_i) = z$)), $\forall z \in \mathcal{Z}$
6: **Initialize** init_queue = Queue(RandomOrder($x_i \in \mathcal{D}$ if $x_i$ is a starting observation))
   **// Start simulation**
7: **Initialize** $\tau = [\ ]$
8: $x = $ init_queue.pop()
9: **for** step $t = 1$ to $\infty$ **do**
10:     $\pi = \mathbb{A}(\tau)$             ▷ update learner with the history
11:     $z = f(x)$
12:     **while** no transition has been accepted for this step **do**
13:         **if** queues[$z$] is empty **then** terminate
14:         Sample a transition $(\cdot, a, r, x', \pi_b(\cdot|x)) = $ queues[$z$].pop()
15:         Calculate $M = \max_{\tilde{a}} \frac{\pi(\tilde{a}|x)}{\pi_b(\tilde{a}|x)}$
16:         Sample $u \sim \text{Unif}(0, 1)$
17:         **if** $u > \frac{\pi(a|x)}{M\pi_b(a|x)}$ **then** reject the sampled transition
18:     $\tau$.append($x, a, r, x'$);  $x = x'$       ▷ update history with new transition
19:     **if** episode ends **then**
20:         **if** init_queue is empty **then** terminate
21:         $x = $ init_queue.pop()
22: **Output:** History $\tau$

---

### A.2.1 Latent State Encoder

Recent work has proposed various ways of learning the latent state encoder for block MDPs, including (Du et al., 2019), HOMER (Misra et al., 2020), PPE (Efroni et al., 2021), AC-State (Lamb et al., 2022), AIS (Subramanian et al., 2022). In this work, we adapt the approach used in HOMER, which can learn a latent state abstraction for kinematic inseparability (Misra et al., 2020), where two observations are combined into a single latent state if and only if (i) they have the same distribution over next observations, and (ii) they have the same (joint) distribution over previous observations and previous actions. Instead of the online interactive version of the HOMER algorithm, we take the state abstraction learning component and train it on offline data. As proven by Misra et al. (2020), the kinematic inseparability abstraction can be learned via a contrastive estimation objective in a supervised classification problem:

$$\arg\min_{p,\phi} \mathbb{E}_{(x,a,x',y)\sim\tilde{\mathcal{D}}}[-\ln p(y \mid \phi(x), a, \phi(x'))]$$

where $\phi$ is the latent state encoder that takes a single observation as input, $p$ is a binary classifier for a transition $\phi(x), a, \phi(x')$ in the latent space, and the overall objective is minimizing the negative likelihood of a binary classification problem. The noise contrastive distribution $\tilde{\mathcal{D}}$ is constructed as follows: first randomly draw a transition $(x, a, x') \in \mathcal{D}$ from the logged dataset, then either $(x, a, x', y = 1)$ or $(x, a, \tilde{x}', y = 0)$ is produced with probability $1/2$, where $\tilde{x}'$ is from an independent draw from the logged dataset $(\tilde{x}, \tilde{a}, \tilde{x}') \in \mathcal{D}$, and $y$ indicates whether the transition is real (1) or an imposter (0). Instead of preconstructing the distribution $\tilde{\mathcal{D}}$ into a full dataset, for efficient implementation during mini-batch learning, we independently sample twice from the logged dataset and include both $(x, a, x', y = 1)$ and $(x, a, \tilde{x}', y = 0)$ in a batch. For more details on the theoretical guarantees, see Misra et al. (2020).

## B Experimental Details on Grid-World

**Environment description**  We consider a navigation task in a grid world (Figure 6), based on Zintgraf et al. (2019). The discrete state space is represented by the index of each cell, with $|\mathcal{S}| = 25$. The agent starts from the bottom left cell, and the goal is the top right cell. There are 5 actions: *stay, up, right, down, left* that each deterministically moves the agent to the adjacent cell in that direction (if the action would move agent out of bounds then the agent stays at the current cell). Reward is $+1$ for reaching the goal and $-0.1$ for each step otherwise. There is no cap on maximum episode length.

| 20 | 21 | 22 | 23 | **G** |
|----|----|----|----|----|
| 15 | 16 | 17 | 18 | 19 |
| 10 | 11 | 12 | 13 | 14 |
| 5 | 6 | 7 | 8 | 9 |
| **S** | 1 | 2 | 3 | 4 |

$$\mathcal{A} = \{\circ, \uparrow, \rightarrow, \downarrow, \leftarrow\}$$

Figure 6: Grid-world with discrete observations, the latent state space and the action space.

### B.1 Illustrative Example of Evaluation Protocol (Figure 2)

We consider an observation space $\mathcal{X} = \mathcal{S} \times \{0 \dots C - 1\}$ induced by the emission function $x = q(s) = s + c|\mathcal{S}|$, where an observation is the latent state $s$ augmented by a modulo counter $c$. At every transition, the counter value is incremented by 1 and then modulo $C$, i.e. $c_{t+1} = (c_t + 1) \mod C$. The initial observation samples the counter $c$ uniformly. This setup is motivated by a common way to construct observations in real-life applications where some cyclical/periodic element is included, such as time of day.

We collected 1,000 episodes using a uniformly random policy in this environment with $C = 32$ to create the offline dataset, corresponding to $134,925$ transitions. For simulating learners, we used a model-based simulation strategy, which first builds an environment model via maximum likelihood estimation of the transition dynamics (assuming the reward function is given), and then uses the estimated environment model as the simulator to interact with the agent. We considered two versions of model-based simulation: sim1 builds the environment model in the observation space $p(x'|x, a)$, and sim2 does so in the latent state space $p(s'|s, a)$ (assuming $s$ is recorded in the data) and uses a uniform emission function $q(x|s) = 1/|C|$ for $x = [s, c]$ where $c \in \{0 \dots C - 1\}$.

For Figure 2, we used the model-based simulations described above for simulating Q-learning with $\epsilon$-greedy exploration using the following hyperparameters: exploration rate $\epsilon = 0.9$, learning rate $\alpha = 0.5$, discount $\gamma = 0.95$. The model-based simulation is repeated for 100 runs using different random seeds, and the results are averaged. This is compared to online Q-learning using the same hyperparameters (also averaged over 100 repeated runs). To illustrate the trade-off between fidelity and efficiency, we truncated the simulation with observation-space model to 50,000 training steps, and the simulation with latent state model to 100,000 steps. We see from Figure 2 that using the observation-space model is not a high-fidelity simulation, since its learning curve starts to deviate from online Q-learning after around 20,000 training steps.

## B.2 Grid-World with Discrete Observations

For this experiment, we use a discrete observation space $\mathcal{X} = \mathcal{S} \times \{0, 1\}^k$ induced by the emission function $x = q(s) = s + (b_1 + \cdots + b_k 2^{k-1})|\mathcal{S}|$, where an observation is made up of the underlying state $s$ and $k$ noisy bits $[b_1 \ldots b_k]$ that are stochastically sampled at every time step. For numerical study we considered $k = 1, 2, 3, 4$, corresponding to observation spaces that are $2, 4, 8, 16$ times larger than the state space. For illustration, we show $k = 4$ which has the clearest trend.

We first collected 1,000 episodes following a uniformly random policy, and then used PSRS on the logged dataset to simulate different "learners". Depending on the type of learner, we used different $g$ function to measure the simulation fidelity. We compared PSRS with the observations $x$ and with the underlying states, and in each case, we repeated the simulation procedure for 100 runs with different random seeds. Both simulations are compared to the learner interacting with the real environment, also for 100 repeated runs.

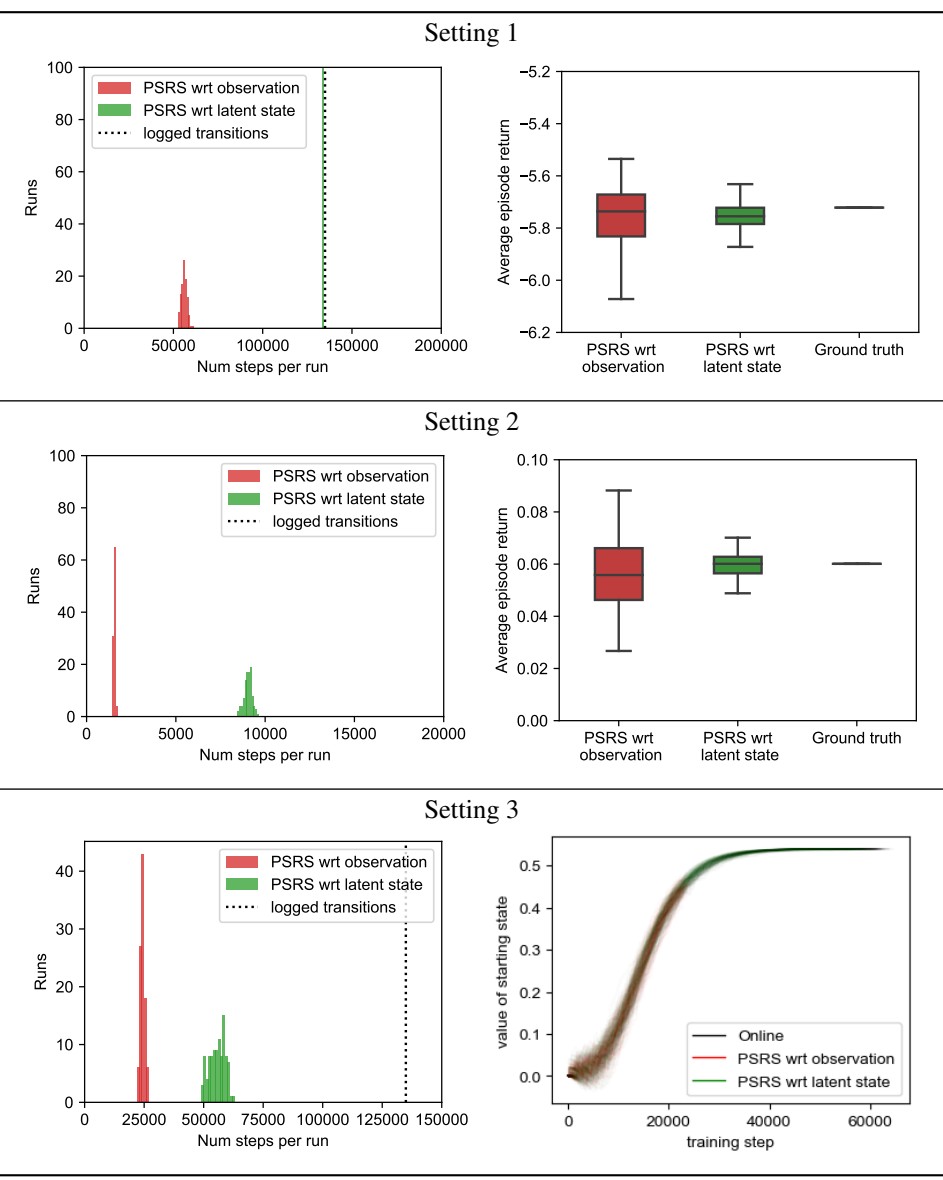

Table 1: Grid-world, observation = latent state + 4 random bits.

**Setting 1: Agent performs Monte-Carlo evaluation of a uniformly random policy** ($\gamma = 0.99$)**.**
Here, instead of tracking the progress over many steps, we use the final MC policy value estimate as
the $g$ function. Because the evaluated policy is identical to the logging policy, the sampled transition
in PSRS is never rejected, and PSRS with latent states can make use of all the samples. However,
compared to using the observations, using latent states led to simulations that are roughly two times
longer, and produce a tighter confidence interval in the resulting value estimate. Both are unbiased
estimate of the groud-truth policy value.

**Setting 2: Agent performs Monte-Carlo evaluation of a near-optimal policy** ($\gamma = 0.99$)**.** We
first learned the optimal Q-function on the true environment, and then constructed an $\epsilon$-soft policy
with $\epsilon = 0.2$. We used the same $g$ function as in Setting 1. Because the evaluated learner's policy
is different from the logging policy, some of the sampled transition in PSRS are rejected, and the
simulations are much shorter (about $1/10$ of Setting 1). The general trend in Setting 1 still holds.

**Setting 3: Agent performs Q-learning with $\epsilon$-greedy exploration** ($\gamma = 0.95$)**.** This setting is
discussed in Figure 3. Here, the learner is a tabular Q-learning agent with zero initialization, and uses
an $\epsilon$-greedy exploration with constant exploration rate of $\epsilon = 0.9$.

**Discussion**   For the setting with a discrete observation space, we presented results for the sim-
plest setting where the observation is the latent state augmented with independently sampled noise
bits. Future work should consider additional variations, such as observations containing a counter
that increments at each step (and modulo a maximum number), or observations made up of the
history of previous latent states and previous actions. For these settings, and we hypothesize further
modifications to PSRS are necessary to enable accurate and efficient offline simulations.

## B.3 Grid world with Continuous Observations

We considered an observation space $\mathcal{X} = [0,1]^2$, induced by the emission function $x = q(s) = [(s\%5) + u_x, (s//5) + u_y]$ where the noise values $u_x, u_y \sim \text{Unif}(0, 0.2)$ are randomly drawn for each time step. Each observation $x$ is a 2D Cartesian coordinate in the unit box where the entire $5 \times 5$ grid-world lies in. We collected the logged dataset containing 1,000,000 transitions using a uniformly random policy, for which all recorded observations are visualized in Figure 7-left. Because the behavior policy takes each of the five actions (essentially a random walk), the state visitation in the logged dataset is more concentrated towards the lower left region around the starting state, and less frequent towards the upper right region around the goal cell.

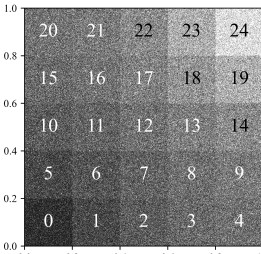 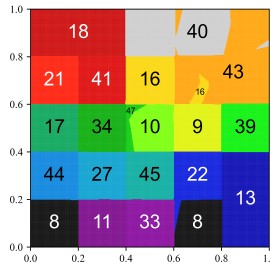

Figure 7: Comparison of the real latent states (left) and learned latent states (right) in the $5 \times 5$ grid world environment. The left figure also shows the state visitation distribution in an offline dataset collected using the random policy.

Since the observation space is now continuous – essentially no two $x$'s are the same – vanilla PSRS on the observation space is no longer practical. Therefore, we need to apply Algorithm 4 instead, which requires a mapping from observation to latent states. For learning the latent state encoder from offline data, we performed a 50-50 split of the logged dataset to create the training and validation sets containing individual transitions. In order to learn discrete latent states for kinematic inseparability abstraction, we used an architecture and the contrastive estimation objective similar to Misra et al. (2020). The encoder network is a one hidden layer neural network with an adjustable hidden layer size and leaky ReLU hidden activation. It applies to both the current observation $x$ and next observation $x'$, mapping the observation to a vector of logits corresponding to the latent dimension. We incorporated both the backward and forward discrete bottlenecks by applying Gumbel-softmax layers to the encoding of both the current and next observations. The contrastive learning objective is a classification task trying to distinguish between observed and imposter transitions $(x, a, x')$; in our implementation, we used another one hidden layer network with the same hidden layer size and activation as the encoder. To implement this training objective, in the training loop, we iterate through two batches of data (shuffled differently): the observed transitions $(x_1, a_1, x_1')$ are used with classification label 1 and the imposter transitions $(x_1, a_1, x_2')$ have classification label 0. We performed a hyperparameter search over the latent dimension, network capacity, learning rate, and different initializations, and selected the final model based on the best validation performance (measured by the classification loss). The final model has a latent dimension of 50, trained with a learning rate of $10^{-3}$ and a hidden layer size of $64$. The learned latent states did not use the full capacity given and are not a perfect match with the ground-truth latent states, as shown in Figure 7; some of them can combine the true latent states (e.g., the learned state 13 combines true states 4 and 9, learned state 8 combines true states 0 and 3, and the states being combined are not necessarily contiguous), or sub-dividing one latent state into multiple (e.g., true state 12 is divided into learned states 10 and 47).

To verify whether the learned latent states are sensible, we conducted a small qualitative experiment comparing the rollouts in the PSRS simulation vs in the real environment when the same sequence of actions is taken by the agent (this is achieved by first performing the PSRS rollouts, and replaying the sequence of actions as generated by PSRS in the real environment). We used a near-optimal $\epsilon$-soft policy with $\epsilon = 0.3$ derived from the optimal Q-function in the real environment. As shown in Figure 8, the PSRS simulation led to a different distribution of trajectories compared to rollouts in the real environment. In particular, because of the learned latent states combining certain non-adjacent states, the simulated trajectories are likely to jump from the starting state to the lower right region, which does not happen in the real environment. For example, in Figure 9-left, the simulated trajectory deviates substantially from the real trajectory, whereas in other cases such as Figure 9 middle and right, the simulated trajectory matches well with what would be observed in the real environment.

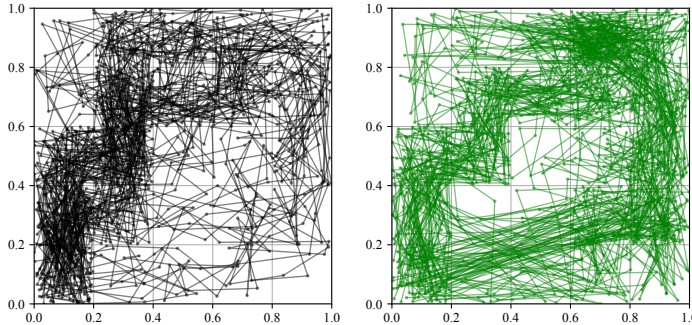

Figure 8: Left: real environment rollouts. Right: PSRS simulation rollouts.

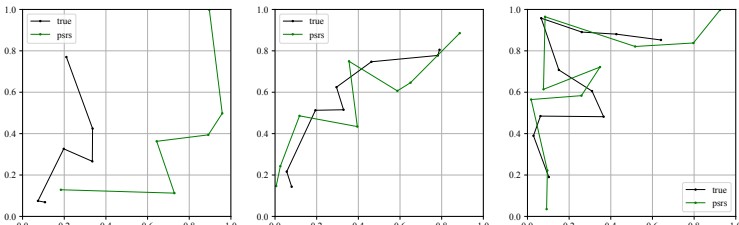

Figure 9: Three comparisons of rollouts in the real environment and from PSRS simulation.

We subsequently used the learned state encoder in offline simulation of a PPO agent. We based our PPO implementation on a public repository,[2] and used the following hyperparameters: discount factor of 0.99, the actor-critic network has two hidden layers each containing 32 units and ReLU activation, training up to 50 epochs with 5000 steps per epoch. We track two quantities as the $g$ function:

- the learning performance, i.e., average episode returns of the learning agent within each training epoch (shown in Figure 5-left)
- the validation performance at the end of every epoch as the $g$ function, estimated by the average returns of the policy at that time from 10 full episodes (Figure 5-right, Figure 10, Figure 12-left). We also report average episode lengths, which is inversely correlated with episode returns (Figure 11, Figure 12-right).

In addition, we compared to the following baselines:

- PSRS-oracle: in place of the learned encoder, an oracle encoder is used that maps the observations back to their ground-truth latent states
- PSRS obs only: within PSRS, ignores the action probabilities (i.e., ignores the rejecting sampling ratio) and just accepts the transition from the queue of the corresponding state
- PSRS act only: within PSRS, ignores the per-state queues and randomly draw a transition, but respects the action probabilities and rejection sampling ratio
- PSRS random: randomly draws a next observation from all observed transitions, ignoring both the state and action

Each simulation setting is repeated for 10 runs, for which the results are averaged. We also performed 10 runs of the PPO agent in the real environment. We visualized the learning curves of validation episode returns and validation episode lengths of each simulation (and the real run) in Figures 10 and 11, and summarize the average results in Figure 12. Despite the errors in the latent states from the learned encoder, it performs close to the oracle encoder, and both are close to the real online PPO in the real environment. All other baselines, despite being efficient, are far from accurate simulations and have non-negligible error. Overall, this experiment shows promise that for block MDP, we can learn to encode the observations into latent states and then do offline simulation in an efficient and accurate manner, completely using offline data.

---

[2]https://github.com/openai/spinningup/tree/master/spinup/algos/pytorch/ppo

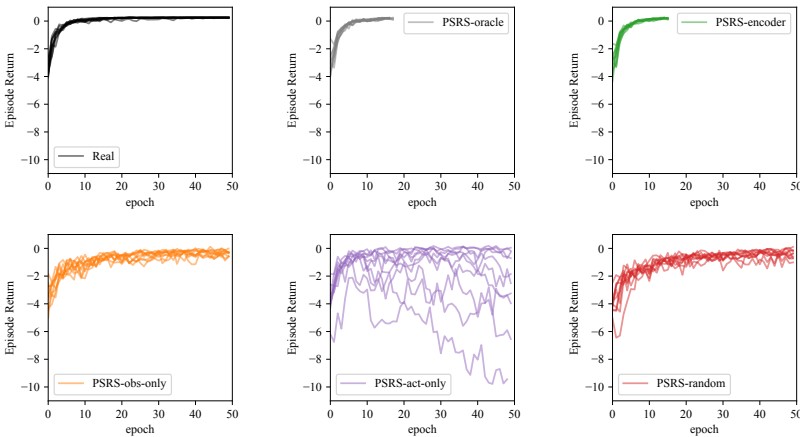

Figure 10: Individual learning curves from each simulation and each of the 10 runs, measured as validation policy performance.

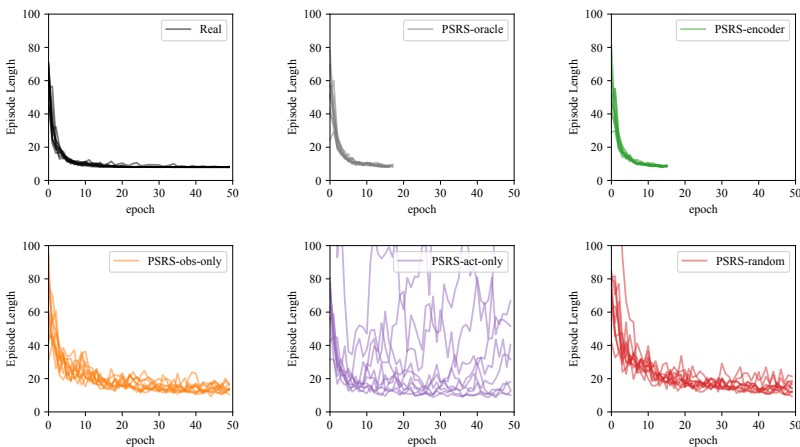

Figure 11: Individual learning curves from each simulation and each of the 10 runs, measured as validation episode lengths.

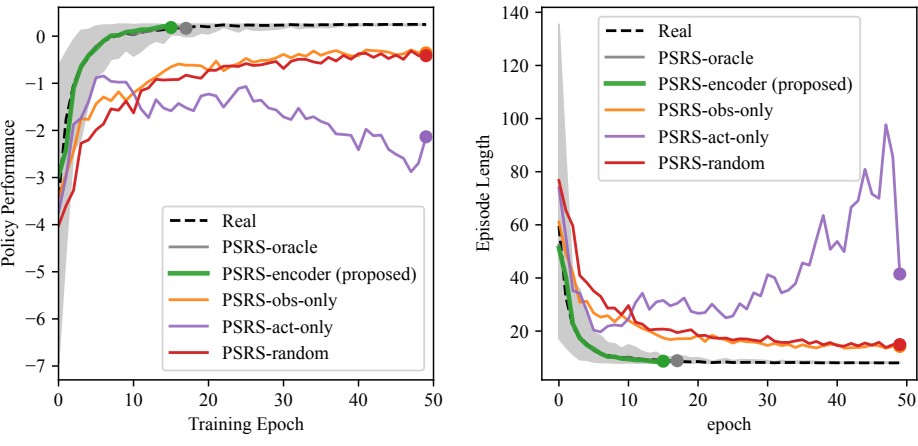

Figure 12: Average learning curves in terms of validation policy performance and episode lengths.

## B.4 Simulating Exo-MDP

Here, we revisit the setting considered in Appendix B.1 and apply our proposed PSRS-based simulation. Recall that in this problem, we have an observation space $\mathcal{X} = \mathcal{S} \times \{0 \ldots C - 1\}$ induced by the emission function $x = q(s) = s + c|\mathcal{S}|$, where an observation is the latent state $s$ augmented by a modulo counter $c$. At every transition, the counter value is incremented by $1$ and then modulo $C$, i.e. $c_{t+1} = (c_t + 1) \mod C$. The initial observation samples the counter $c$ uniformly. This setup is motivated by a common way to construct observations in real-life applications where some cyclical/periodic element is included, such as time of day.

We collected 1,000 episodes using a uniformly random policy in this environment with $C = 32$ to create the offline dataset, corresponding to $134,925$ transitions. We then applied PSRS on the logged dataset to simulate tabular Q-learning. We compared PSRS with the observations $x$ vs PSRS with the underlying states $s$, and in each case, we repeated the simulation procedure for 100 runs with different random seeds. As expected, PSRS with the latent states (green) is more efficient than using the raw observations (red) and led to longer simulations (Figure 13). However, perhaps surprisingly, PSRS using the latent states is not an accurate simulation. What makes this problem different from the one considered in Appendix B.2? In both problems (grid-world with noise vs grid-world with modulo counter), the underlying grid-world MDP is identical; so are the (presumed) latent state spaces and the controllable latent dynamics.

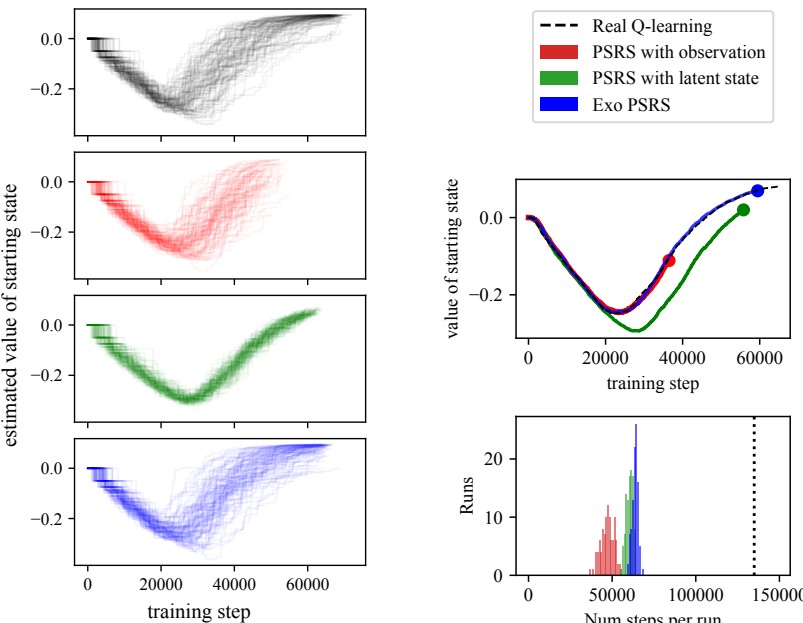

Figure 13: Grid-world with discrete observations. Left: estimated value of starting state for real Q-learning and two simulations (using observations and using latent states directly). Right: fidelity and efficiency of the simulations. Both simulations have perfect fidelity, but using latent state allows for simulating longer.

The main difference comes from the transition dynamics in the observation space. For the grid-world with noise, since we draw the noise bits randomly at each step, the exogenous noise is time-independent and we can write $p(x'|x,a) = p(s',c'|s,c,a) = p(s'|s,a) \, p(c')$. However, for the grid-world with modulo counter, the "noise" is time-dependent and forms an exogenous process, where $p(s',c'|s,c,a) = p(s'|s,a) \, p(c'|c)$ where $p(\cdot|c_1) \neq p(\cdot|c_2)$ for some $c_1 \neq c_2$. When we apply Algorithm 4 using the latent states $s$, we are destroying the exogenous process in the counter $c$, resulting in the simulation producing impossible transitions, e.g. $c = 1$ transitions to $c' = 20$. Since the learner we are simulating operates in the raw observation space, such implausible transitions will lead to inaccurate simulations.

Here, we provide one solution for simulating these types of exogenous MDPs. We assume access to the following oracles: $f : \mathcal{X} \to \mathcal{Z} \times \mathcal{C}$ that "splits" an observation into its endogenous part $z$ and exogenous part $c$, $f^{-1} : \mathcal{Z} \times \mathcal{C} \to \mathcal{X}$ that does the inverse and "merges" the endogenous and exogenous parts into an observation, and reward oracle $g$. We modify the latent PSRS Algorithm 4 to account for exogenous processes as shown in Algorithm 5, where we separately maintain a queue of

the exogenous state $c$ and maintain the dynamics of the exogenous process in the simulation using this queue. We highlight the main algorithmic modifications in magenta. Applying Algorithm 5 to simulate Q-learning on the problem (blue approach in Figure 13) leads to both more efficient and more accurate simulations, compared to the two approaches considered earlier in this section.

---

**Algorithm 5** Latent PSRS simulation for Exo-MDP

---

1: **Input:** Logged dataset $\mathcal{D} = \{(x_i, a_i, r_i, x_i', \pi_i(\cdot|x_i))\}$ where $a_i \sim \pi_i(\cdot|x_i)$
2: **Input:** Learner $\mathbb{A}$ that maps from history $\tau$ to policy $\pi$
3: **Input:** Oracles $(z, c) = f(x)$ and $x = f^{-1}(z, c)$
4: **Input:** $g(z, c)$, expected reward for a transition reaching $x = (z, c)$
5: **Preprocess** for each $i$, calculate $(z_i, c_i) = f(x_i)$, $(z_i', c_i') = f(x_i')$
6: **Initialize** $\forall z \in \mathcal{S}$, zqueues[z] = Queue(RandomOrder($(z_i, a_i, z_i', \pi_i) \in \mathcal{D}$ s.t. $z_i = z$))
7: **Initialize** $\forall c \in \mathcal{C}$, cqueues[c] = Queue(RandomOrder($(c_i, c_i') \in \mathcal{D}$ s.t. $c_i = c$))
8: **Initialize** init_queue = Queue(RandomOrder($x_i \in \mathcal{D}$ if $x_i$ is a starting observation))
   // Start simulation
9: **Initialize** $\tau = [\ ]$
10: $x = $ init_queue.pop()
11: **for** step $t = 1$ to $\infty$ **do**
12:     $\pi = \mathbb{A}(\tau)$                                       ▷ update learner with the history
13:     Calculate $(c, z) = f(x)$
14:     **while** no transition has been accepted for this step **do**
15:         **if** zqueues[z] is empty or cqueues[c] is empty **then** terminate
16:         Sample an endogenous transition $(z, a, z', \pi) = $ zqueues[z].pop()
17:         Calculate $M = \max_{\tilde{a}} \frac{\pi_{\mathbb{A}}(\tilde{a}|x)}{\pi(\tilde{a}|x)}$
18:         Sample $u \sim \text{Unif}(0, 1)$
19:         **if** $u > \frac{\pi_{\mathbb{A}}(a|x)}{M\pi(a|x)}$ **then** reject the sampled transition
20:     Sample an exogenous transition $(c, c') = $ cqueues[c].pop()
21:     Calculate $x' = f^{-1}(z', c')$, $r = g(z', c')$
22:     $\tau$.append($x, a, r, x'$);   $x = x'$                      ▷ update history with new transition
23:     **if** episode ends **then**
24:         **if** init_queue is empty **then** terminate
25:         $x = $ init_queue.pop()
26: **Output:** History $\tau$

---

Note that the experiments above only show what might be required for efficient and accurate simulations for Exo-MDPs and we imposed strong assumptions on the oracles. In practice, such oracles are rarely given directly and must be learned from data, and we believe this is an important area of future research.

Moreover, these experiments highlight an important consideration for choosing the type of latent state discovery algorithms. For example, the AC-State formulation proposed in Lamb et al. (2022) explicitly extracts the agent-controllable latent states while removing any exogenous noise or processes (in their example where the observation is an image of a scene containing a robot arm, the video played on TV screen in the background is not controllable and thus exogenous). Applying latent PSRS (Algorithm 4) using the latent states discovered by AC-State would lead to inaccurate simulations, e.g. the simulated observation sequence would not be a coherent video on the TV screen. We note that this may be acceptable as long as the simulated learner ignores background. However, if exogenous states (though uncontrollable) are important to the learner otherwise (e.g. it specifies the reward) then such simulations are not reliable.

