# OpenReview forum: "Towards Data-Driven Offline Simulations for Online Reinforcement Learning"
_NeurIPS.cc/2022/Workshop/Offline_RL — Offline RL Workshop NeurIPS 2022_

### Official Review · Reviewer_QZBP · 2022-10-14

**Rating:** 6
**Confidence:** 4

**Review:**

The paper's focus is on creating a simulated environment from an offline dataset of interactions, that can be used to train an online RL learning system. One way of doing this would be model based RL where a model is fit based on the data. This paper aims for a different approach - replaying data that we've seen before whenever we do a transition.

The baseline compared to is PSRS, per-state rejection sampling. Given a dataset of $(s,a,r,s')$, we first preprocess into a hashtable of $s \to \{(a, r, s')\}$, mapping state + action to a list of examples observed offline. Whenever the model is in state $s$, we pop example $(a, r, s')$ from the queue, and accept / reject based on similarity of the action distribution and behavior policy distribution. If we run out of examples for an $s$, the simulator stops. When rejection sampling is done correctly, this is an unbiased estimator of experience that allows for longer simulation than creating a hashtable of $(s,a) \to \{(r,s')\}$ (since there will be times where we can use a slightly different action $a$ without affect the trajectory distribution in expectation).

This paper is, essentially, an extension of PSRS into continuous MDPs. The authors propose first learning a latent state encoder that maps continuous state to a discrete latent vector. Then, PSRS is done over the discrete latent state. The state encoding is done with the HOMER algorithm of Misra et al 2020, which aims to cluster states based on their similarity within an RL context.

The paper is clear and the combination of these latent state encoders with PSRS makes a great deal of sense, but I was somewhat let down that the benchmarked environments are:

* A discrete grid world with 16x as many states as the original one (sample 4 random bits per timestep)
* A 2d continuous grid world navigation task with no walls.

These settings seemed fairly small to me relative to other offline RL tasks. It does not seem hard to learn a latent state encoder in these environments, and the method's success basically rides entirely on how good the latent state encoder is.

Also, the latent state encoder and PSRS are both from prior work, this work is just observing that you can combine the two. The learning of HOMER is all done prior to using it in PSRS, which is also done conventionally.

Overall I am not too excited about the paper but I do not have any specific fault with it, the paper is well-written and does exactly what it said it would do.

---

### Official Review · Reviewer_ukJb · 2022-10-18
**A semi-parametric method for offline learner simulation.**

**Rating:** 6
**Confidence:** 4

**Review:**

Offline learner simulation is an important problem when the real environment is not accessible. This paper proposed a semi-parametric method for offline learner simulation.  It is built on top of per-state rejection sampling method by grouping the unobserved latent states. It further proposed two evaluation metrics for simulation, i.e., efficiency and fidelity. Experiments on the grid world navigation task demonstrate promising results.

In general, this paper is well written. It proposed a simple method for offline simulation on top of the classical per-state rejection sampling method. The main concern is in the grid world navigation simulation task. The main motivation of this paper is to facilitate offline simulation on those tasks where online reinforcement learning is expensive, such as recommendation and robot control. However, the experiment is about a grid word navigation task, which does not handle this challenge. It is better to demonstrate the offline simulation performance on more real tasks.